# Seafarers’ Physical Activity and Sleep Patterns: Results from Asia-Pacific Sea Routes

**DOI:** 10.3390/ijerph17197266

**Published:** 2020-10-05

**Authors:** Ik-Hyun Youn, Jung-Min Lee

**Affiliations:** 1Division of Navigation and Information Systems, Mokpo National Maritime University, Mokpo 58628, Korea; iyoun@mmu.ac.kr; 2Department of Physical Education, Kyung Hee University, Yongin 17104, Korea

**Keywords:** seafarer population, physical health, physical activity, sleep pattern, wearable sensor

## Abstract

Prolonged ocean voyages constrain the regular physical activity and sleep patterns of seafarers. However, there is a lack of information on seafarers’ physical activity and sleep behavior. The purpose of this study was to systematically examine physical activity and sleep patterns among seafarers using a single wrist-worn accelerometer. Fifty-one senior maritime students (mean age = 22.8 years; 80.0% male) in a university navigation department participated in the study. Data were collected from participants on three sea voyages in the Asia-Pacific region. Indicators of moderate to vigorous intensity physical activity (MVPA) and sleep patterns were compared between several conditions: (1) moored versus sailing, (2) on-navigation duty and off- navigation duty, and (3) day versus night navigation duty. Regardless of conditions, low levels of physical activity and short sleep durations were observed. Independent sample t-tests revealed that time spent doing MVPA was significantly higher when participants were off-duty than when they were on-duty (*p* < 0.001). Physical activity did not significantly differ between the other two conditions. While total sleep duration was not significantly different between mooring and sailing, the results showed that participants awakened more frequently (*p* = 0.007) and their sleep was more restless (*p* < 0.001) while sailing. The results demonstrated that developing effective programs to promote physical activity should be a public health priority for the seafaring population, and serious consideration is required to mitigate sleep disruption during sailing.

## 1. Introduction

Seafarers often work in unstable and confined living environments for several months at a time. Since the unique living conditions at sea require seafarers to exert more physical and neurological effort to maintain regular physical conditions (e.g., movement, position, and balance) [1,2], significant lifestyle changes and health risks in the seafaring population have been reported. For example, Hjarnoe and Leppin examined health risks among Danish seafarers and reported that significant lifestyle changes were associated with health hazard factors, such as a low level of fitness and metabolic syndrome [3]. Moreover, Scovill and colleagues investigated lifestyle-related cardio-metabolic disease among seafarers on vessels in United States’ inland waterways and identified various chronic disease risk factors, such as a high prevalence of physical inactivity and obesity [4]. Research by Kamada also found that crew members aboard a ship complained of neurotic symptoms and chronic fatigue [5].

Though these studies have contributed valuable knowledge on the health aspects of the maritime population, the outcomes have primarily relied on subjective measures (e.g., interviewing and self-report questionnaires) to quantify physical activity and sleep patterns. Few unobstructed, objective physical activity and sleep measurement studies have been conducted to assess seafarers’ key health behaviors. In this study, a wearable sensor was utilized to objectively assess seafarers’ physical activity and sleep patterns in the unique living conditions at sea. Since wearable sensors are advantageous over subjective measures in terms of feasibility, validity, and reliability, wearable sensors have been widely utilized in various physical activity and sleep assessment studies [6,7,8,9]. 

Therefore, the purpose of this study was to compare the intensity of physical activity and the sleep patterns of seafarers in the following three conditions: (1) mooring versus sailing, (2) on-navigation duty and off- navigation duty, and (3) day versus night navigation duty. Twenty-four-day voyages across sea routes in South Korea, Singapore, and Vietnam were selected to represent typical maritime navigation conditions of open sea voyages in the Asia-Pacific region. 

## 2. Materials and Methods 

### 2.1. Study Population

The study protocol was approved in advance by the Institutional Review Board (IRB) at the University of Nebraska Medical Center. The principal investigator used a recruitment flyer and word-of-mouth to invite participants. All participants (N = 51; 10 females and 41 males) were senior apprentices on one of two training ships from the Mokpo National Maritime University in South Korea. All participants were students in the navigation department. Volunteer apprentices from the engineering department were not eligible to participate. Before initiating the study, self-reported age (22.8 ± 2.0 years), weight, and height were obtained (Table 1). As a minimum requirement for onboard training, all trainee have to officially report their medical condition to the government authority. Based on the medical condition report, it was confirmed that the participants for the study did not have any pre-existing medical condition as navigation officers.

All participants were made aware of the procedures and purpose of the research before they signed the informed consent form (IRB No.: 273-16-EP). 

### 2.2. Data and Methods

A single wrist-worn accelerometer was used to collect data on participants’ physical activity and sleep patterns for four consecutive days under both mooring and sailing conditions. Participants were asked to wear a single ActiGraph GT9X Link (ActiGraph, Pensacola, FL, USA) attached to the dominant wrist with an elastic strap. The sensor uses a triaxial accelerometer and is appropriate to collect data on seafarers’ activity aboard a ship because of its light weight (i.e., 14 grams), memory space (i.e., four gigabytes), and high water resistance. The ActiGraph GT9X Link has been extensively validated with diverse samples in physical activity [9,10,11] and sleep research [12,13,14]. The wearable sensor measures both gravity and wrist movement, and the sampling frequency was set to 100 Hz, which was similar to the frequency employed in previous studies [15,16]. 

The physical activity level was estimated using the GGIR R-package (GGIR) [17]. GGIR is freely available and enables the processing and analysis of three-dimensional acceleration data. The signal processing includes automatic calibration, the detection of abnormally high acceleration values, the detection of non-wear, and the calculation of the average magnitude of dynamic acceleration [17,18]. The outcome parameters were used to calculate moderate to vigorous intensity physical activity (MVPA) in one-minute intervals per 24-h period. Within each condition, the mean number of MVPA minutes over a 24-h period was calculated. 

ActiLife software (ActiGraph, Pensacola, FL, USA) was used to estimate sleep variables from the raw data using the validated Sadeh algorithm [19,20]. The acceleration data were processed in one-minute intervals. The sleep variables included (1) the total sleep time (TST), (2) the number of awakenings (NoA), and (3) the sleep fragmentation index (SFI). The TST was measured as the total sleep duration in minutes, from falling asleep to awakening. The NoA was the number of awakenings lasting more than five minutes. The SFI represents an index of restlessness during the TST, described as a percentage [21,22]. The higher the SFI value, the poorer the sleep quality. 

The annual sailing plan of the training ship that was employed in the study contained sixteen voyages per year. The annual sailing plan consisted of fourteen coastal voyages (i.e., sailing between Korean ports) and two international voyages. The experiment was conducted during the second semester, and experiment participants had experienced twelve domestic voyages and one international voyage before the experiment. Data were collected from participants on three sea voyages: (1) from the Port of Mokpo in South Korea to the Port of Singapore in Singapore (N = 23), (2) from the Port of Singapore to the Port of Da Nang in Vietnam (N = 11), and (3) from the Port of Da Nang to the Port of Mokpo (N = 17). At the beginning of each voyage, participants were asked to wear the wearable sensor at least for four consecutive days except while bathing. The sea routes, represented in Figure 1, totaled approximately 5100 nautical miles passing through the East China Sea, South China Sea, Singapore Strait, and the Gulf of Thailand. 

The selected sea voyage routes are representative of sea routes in today’s maritime shipping industry. More than half of the annual merchant fleet tonnage worldwide and one-third of maritime traffic worldwide passes through these sea routes [23]. Therefore, the ships’ navigation conditions in this study were suitable for investigating seafarers’ physical activity and sleep patterns in merchant marine vessels. The ships’ sailing plans included about 15 days of sailing and 9 days of mooring at ports in both Singapore and Da Nang, Vietnam.

### 2.3. Navigation Watch Schedule

Many shipping companies in Asian countries have adopted the traditional navigation watch system, which consists of three watch teams that each work two four-hour shifts over a 24-h period. The system includes a chief officer watch (i.e., 04:00–08:00 and 16:00–20:00), a second officer watch (i.e., 24:00–04:00 and 12:00–16:00), and a third officer watch (i.e., 08:00–12:00 and 20:00–24:00). In this study, the navigation watch system was also categorized into the day- and night-watch teams according to dominant sleeping times. The chief officer watch and the third officer watch teams comprised the day-watch teams, and the second officer watch comprised the night-watch teams. The day-watch teams maintain their regular daytime activity patterns during sea voyages. The night-watch team, however, must rapidly change its activity patterns at the beginning of a sea voyage, because their main navigation time is from 24:00 to 04:00 and the team needs to sleep between early morning and noon. Figure 2 illustrate work and living environment of navigators onboard. Navigation watch is not required when the ship remains moored at the harbor, so the navigation watch teams are assigned to eight hours of daytime work.

### 2.4. Statistical Analysis

Means and standard deviations (SD) were calculated to summarize the demographic characteristics, physical activity, and sleep variables of participants. A series of independent-samples t-tests were performed to examine the differences in mean physical activity and sleep variables across various conditions: (1) sailing status (i.e., mooring versus sailing), (2) navigation watch status (i.e., on-navigation duty and off-navigation duty), (3) and watch schedules (i.e., day-watch versus night-watch). An additional paired-samples t-test was carried out to compare physical activity levels between navigation watch schedules. All *p*-values were calculated while assuming a two-tailed hypothesis. Statistical analyses were performed using IBM SPSS version 25 (Chicago, IL, USA), and *p*-values of less than 0.05 were considered statistically significant. 

## 3. Results

Overall, physical activity levels were found to be low regardless of the conditions. Table 2 presents the results of two separate independent-samples t-tests: 1) comparing MVPA (minutes/24-h s) between mooring and sailing and 2) comparing MVPA between the day-watch and night-watch teams. 

MVPA did not significantly differ (t (49) = 1.78; *p* = 0.081) between mooring (Mean (M) = 12.46; SD = 4.48) and sailing (M = 15.55; SD = 6.40). Similarly, MVPA for the day-watch teams (M = 17.92; SD = 6.86) was not significantly different (t (32) = 1.83; *p* = 0.077) from the night-watch team (M = 13.95; SD = 5.70). However, MVPA was significantly (t (33) = 5.28; *p* < 0.001) higher while participants were off-duty (M = 9.78; SD = 4.51) compared to on-duty (M = 6.00; SD = 2.83). 

Regardless of conditions, the TST measurements revealed that sleep deprivation was a regular part of a seafarer’s routine. Moreover, based on NoA and SFI scores, it was found that sleep quality during sailing was worse than mooring conditions. The results of the independent samples t-tests comparing the TST, the NoA, and the SFI between mooring and sailing conditions are presented in Table 3. 

The TST during sailing (M = 267.77; SD = 108.83) was not significantly different (t (78.91) = −1.63; *p* = 0.107) from the TST during mooring (M = 304.41; SD = 163.37). However, the NoA was significantly higher (t (101.15) = 2.74; *p* = 0.007) during sailing (M = 14.31; SD = 9.60) compared to mooring (M = 10.31; SD = 9.90). The SFI was also significantly (t (238) = 3.56; *p* < 0.001) higher during sailing (M = 22.31; SD = 14.93) compared to mooring (M = 14.56; SD = 14.56). 

The results of the independent samples t-tests comparing the TST, the NoA, and the SFI between day-watch and night-watch conditions are also presented in Table 3. The TST of the night-watch team (M = 248.17; SD = 88.28) was significantly lower (t (57) = −2.03; *p* = 0.047) than that of the day-watch teams (M = 287.05; SD = 123.33). However, the NoA of the night-watch team (M = 12.09; SD = 8.12) and the day-watch teams (M = 14.88; SD = 10.62) were not significantly different (t (57) = −1.59; *p* = 0.118). Similarly, the SFI of the night-watch team (M = 21.00; SD = 13.93) and the day-watch teams (M = 18.44; SD = 12.56) were not significantly different (t (57) = 1.24; *p* = 220).

## 4. Discussion

This study aimed to systematically quantify and evaluate physical activity and sleep behavior among seafarers in Asian maritime settings. The main finding was that most participants showed extremely low levels of physical activity and poor sleep quality, regardless of whether their ship was sailing or moored. Using self-reported physical activity measurements, previous studies have reported insufficient physical activity [3,4] and poor sleeping quality [24,25] among seafarers. The findings of the current study support and extend these earlier findings with objective measurements of physical activity and sleep quality. 

The Centers for Disease Control and Prevention (CDC) in the USA recommend 150 minutes of MVPA per week to maintain physical health [26,27]. The physical activity levels of moored and sailing seafarers’ in the current study were 58.1% and 72.6%, respectively, thus indicating inadequate physical activity according the CDC guidelines. Physical activity, measured as time spent doing MVPA, was not significantly different between moored and sailing conditions even though ships moored the harbor stayed in much calmer sea conditions, with less motion, than ships at sea. This may imply that reduced physical activity for seafarers aboard a ship is not due to unstable environmental conditions. Instead, the confined living space of a ship might be a limiting factor for participating in adequate physical activity on a ship. 

During sailing conditions, different navigation watch schedules (i.e., day vs. night) were not significantly related to physical activity levels. However, when MVPA was separated into MPA and VPA, the night-watch team showed significantly lower VPA levels than the day-watch teams. Physical activity levels, measured as MVPA, during navigation watch on-duty was significantly greater than the physical activity levels in resting periods (i.e., off-duty). This might have been because participants performed their navigation watch routine in a confined space. Conversely, participants may have performed their daily physical activity during their resting time in a relatively less confined area. 

A minimum of seven hours of sleep per day is recommended for healthy adults [28,29]. However, during mooring and sailing conditions, seafarers’ TST measurements were 27.6% and 36.3%, respectively, of this guideline. The quality of sleep showed significant associations with a ship’s sailing status. While the TST was not significantly different between mooring and sailing conditions, participants awakened more frequently and maintained poorer sleep quality during sea voyages. The unstable movement of a ship might influence the quality of sleep; however, this does not explain why seafarers had similar amounts of TST while moored and while sailing. The effects of a ship’s motion could also explain the difference in sleep quality between the day-watch and night-watch teams. The night-watch teams had significantly lower TST levels than the day-watch teams, but both day- and night-watch teams had similar qualities of sleep based on the NoA and the SFI. 

Compared to the day-watch teams, the night-watch team was more vulnerable to low physical activity and poor sleep quality. Therefore, for seafarers whose duty is to navigate during the night, additional consideration is required to improve physical activity levels and sleep quality. 

This study provided new insight into the quantifiable physical activity and sleep quality of seafarers on ocean voyages. To the best of our knowledge, no study to date has examined physical activity and sleep behavior by utilizing an objective measurement tool (i.e., an accelerometer) in the maritime population. Therefore, this study highlights the importance of worksite health promotion programs for the seafaring population. This study was limited to seafarers in a university navigation department and may not be representative of all seafarers. Further investigation into the physical activity and sleep patterns of seafarers in other professional roles, such as engineers, could provide further insight into the physical activity and sleep characteristics of the maritime population. Additionally, sea conditions are unpredictable, and different conditions may have produced different results. Thus, physical activity levels and sleep patterns should be examined under different seasonal sea conditions, such as those found in regions other than the Asia Pacific region. 

## 5. Conclusions

This study aimed to objectively assess the physical activity and sleep patterns of Asian seafarers while aboard a ship. The use of an accelerometer was innovative, and the results revealed that the physical activity levels and sleep quality of seafarers in Asian maritime settings were considerably lower than recommended physical activity and sleep guidelines. The findings suggest that there is an urgent need to improve physical activity and sleep quality among seafarers to reduce negative health outcomes in the seafarer population. This information could be helpful for determining priorities in the development of worksite health promotion programs for those who spend lengthy periods of time working aboard ships.

## Figures and Tables

**Figure 1 ijerph-17-07266-f001:**
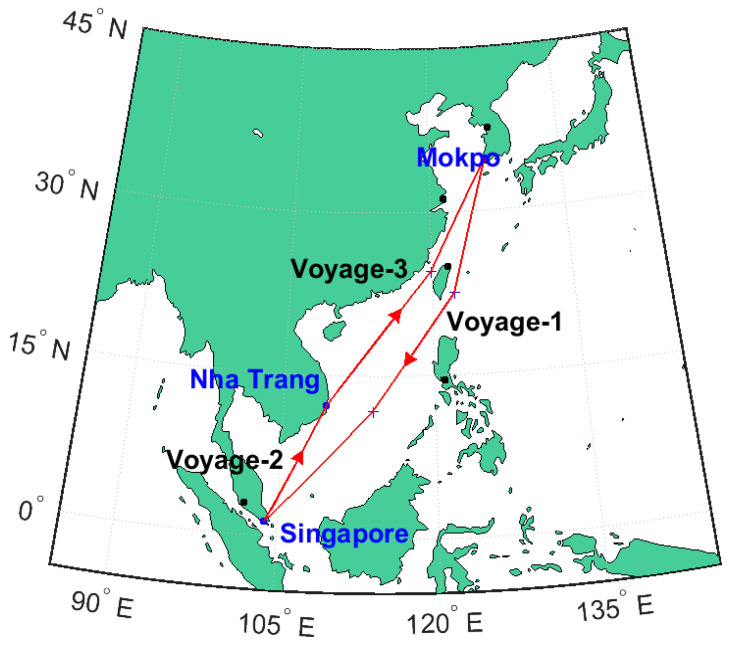
Selected sea routes: Voyage 1: Korea to Singapore; Voyage 2: Singapore to Vietnam; and Voyage 3: Vietnam to Korea.

**Figure 2 ijerph-17-07266-f002:**
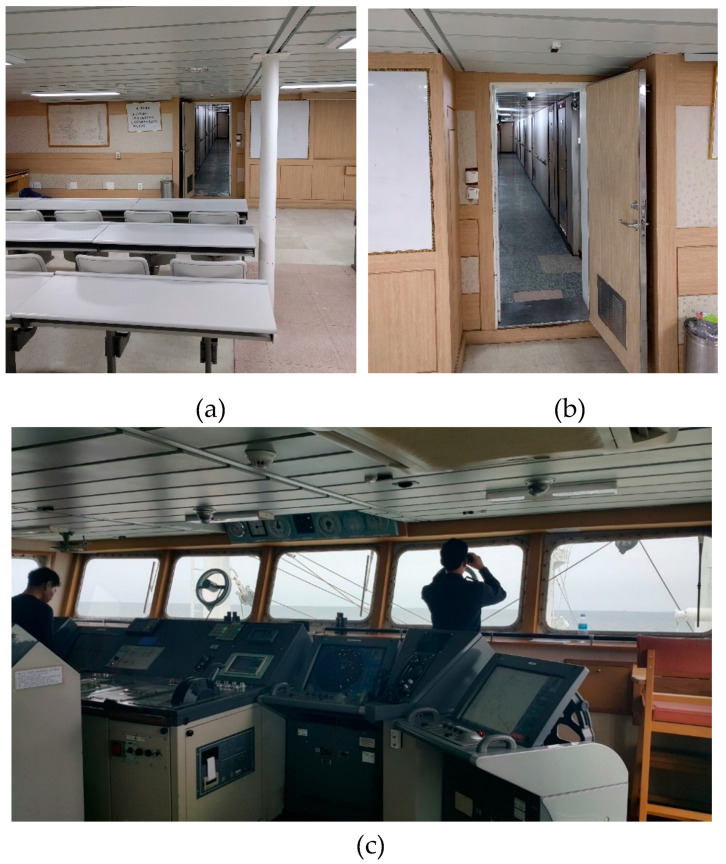
Ship accommodation including lecture room (**a**), hallway (**b**), and navigation bridge (**c**).

**Table 1 ijerph-17-07266-t001:** Descriptive characteristics of participants (N = 51).

Characteristics	Moored (N = 17)	Sailing (N = 34)
	Mean (Standard Deviation)	Mean (Standard Deviation)
Female/Male	3/14	7/27
Age (years)	22.6 (2.1)	23.0 (1.9)
BMI (kg·m^2^)	23.6 (3.5)	24.2 (3.8)
Height (cm)	173.9 (5.6)	176.4 (8.3)
Weight (kg)	70.6 (8.5)	72.6 (11.8)

BMI = body mass index.

**Table 2 ijerph-17-07266-t002:** The comparison of time spent on various physical activity levels in different settings (N = 51).

**Physical Activity Intensity** **(minutes/24 h )**	**Moored (N = 17)**	**Sailing (N = 34)**	***t*** **(*df*)**	***p*** **-Value**
	**Mean (SD)**	**Mean (SD)**		
MVPA	12.46 (4.48)	15.55 (6.40)	1.78 (49)	0.081
MPA	11.99 (4.24)	14.81 (6.04)	1.71 (49)	0. 092
VPA	0.40 (0.37)	0.59 (0.50)	1.42 (49)	0.163
**Physical Activity Intensity** **(minutes/24 h )**	**On-Duty (N = 34)**	**Off-Duty (N = 34)**	***t*** **(*df*)**	***p*** **-Value**
	**Mean (SD)**	**Mean (SD)**		
MVPA	6.00 (2.83)	9.78 (4.51)	5.28 (33)	< 0.001
MPA	5.77 (2.72)	9.25 (4.25)	5.10 (33)	< 0.001
VPA	0.19 (0.15)	0.41 (0.39)	4.11 (33)	< 0.001
**Physical Activity Intensity** **(minutes/ 24 h )**	**Day-Watch (N = 13)**	**Night-Watch (N = 21)**	***t*** **(*df*)**	***p*** **-Value**
	**Mean (SD)**	**Mean (SD)**		
MVPA	17.92 (6.86)	13.95 (5.70)	1.83 (32)	0.077
MPA	16.88 (6.60)	13.38 (5.34)	1.69 (32)	0.100
VPA	0.82 (0.56)	0.44 (0.40)	2.30 (32)	0.028

MVPA = moderate to vigorous physical activity; MPA = moderate physical activity; VPA = vigorous physical activity; and SD = standard deviation.

**Table 3 ijerph-17-07266-t003:** The comparison of sleep time between harbor and sea and between day-watch and night-watch (N = 51).

**Sleep Parameters**	**Moored (N = 17)**	**Sailing (N = 34)**	***t*** **(*df*)**	***p*** **-Value**
	**Mean (SD)**	**Mean (SD)**		
Total Sleep Time	304.41 (163.37)	267.77 (108.83)	−1.63 (78.9)	0.107
Number of Awakenings	10.31 (9.90)	14.31 (9.60)	2.74 (101.2)	0.007
Sleep Fragmentation Index	14.56 (14.04)	22.31 (14.93)	3.56 (238)	< 0.001
**Sleep Parameters**	**Day-Watch (N = 13)**	**Night-Watch (N = 21)**	***t*** **(*df*)**	***p*** **-Value**
	**Mean (SD)**	**Mean (SD)**		
Total Sleep Time	287.05 (123.33)	248.17 (88.28)	−2.03 (57)	0.047
Number of Awakenings	14.88 (10.62)	12.09 (8.12)	−1.59 (57)	0.118
Sleep Fragmentation Index	18.44 (12.56)	21.00 (13.93)	1.24 (57)	0.220

SD = standard deviation.

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
