# Peer review of "Seafarers’ Physical Activity and Sleep Patterns: Results from Asia-Pacific Sea Routes"

_ijerph, 2020, doi:10.3390/ijerph17197266_

Round 1

Reviewer 1 Report

The study offers additional knowledge regarding the relationship between physical activity and sleep patterns of seafarers emplyed in prolonged ocean voyages.

The study is well written and results are of interest.

Since night work has been included among the risk factors in the statistical analysis, the authors should briefly discuss the possible synergistic role of the interaction between physical inactivity and metabolic alterations induced by night work both in altering the sleep pattern and in the induce possible long-term alterations in bone metabolism (Rosa DE, Marot LP, de Mello MT, et al. Shift rotation, circadian misalignment and excessive body weight influence psychomotor performance: a prospective and observational study under real life conditions. Sci Rep. 2019;9(1):19333. Published 2019 Dec 18. doi:10.1038/s41598-019-55114-w; Coppeta L, Papa F, Magrini A. Are Shiftwork and Indoor Work Related to D3 Vitamin Deficiency? A Systematic Review of Current Evidences. J Environ Public Health. 2018;2018:8468742. Published 2018 Sep 10. doi:10.1155/2018/8468742). In recent sistematic reviews, low of vitamin D3 were in fact detected both in night workers and in white collars. 

minor spell check required

Author Response

Reviewer's Notes: Since night work has been included among the risk factors in the statistical analysis, the authors should briefly discuss the possible synergistic role of the interaction between physical inactivity and metabolic alterations induced by night work both in altering the sleep pattern and in the induce possible long-term alterations in bone metabolism. (Rosa DE, Marot LP, de Mello MT, et al. Shift rotation, circadian misalignment and excessive body weight influence psychomotor performance: a prospective and observational study under real life conditions. Sci Rep. 2019;9(1):19333. Published 2019 Dec 18. doi:10.1038/s41598-019-55114-w; Coppeta L, Papa F, Magrini A. Are Shiftwork and Indoor Work Related to D3 Vitamin Deficiency? A Systematic Review of Current Evidences. J Environ Public Health. 2018;2018:8468742. Published 2018 Sep 10. doi:10.1155/2018/8468742). In recent systematic reviews, low of vitamin D3 were in fact detected both in night workers and in white collars. 

Authors’ Response: Thanks for your comment. Based on the relevant literature (Xia, Song-Yun, et al. "Psychological and physiological effects of a long voyage on female seamen in China." Int J Clin Exp Med 9.4 (2016): 7314-7321; Hjarnoe, Lulu, and Anja Leppin. "Health promotion in the Danish maritime setting: challenges and possibilities for changing lifestyle behavior and health among seafarers." BMC public health 13.1 (2013): 1165..), it was revealed that chronic psychological stress during a long voyage effect on the mariner’s psychological and physiological behavior. As our study was intended to compare the physical aspect of the mariner’s health, the submitted paper was not mentioned about the effect of behavior and mood. Our Next study would be examined the psychological behavior based on the length of the voyage.  

Best regards,

Reviewer 2 Report

Here authors have used wearable sensors to assess the physical activity and sleep patterns in seafarers. The sensors are more feasible, reliable, and can be easily validated. They have analyzed physical activity and sleep patterns in seafarers during mooring versus sailing, on and off-duty navigation, and day versus night navigation. They have chosen twenty-four-day open sea voyages across sea routes in South Korea, Singapore, and Vietnam. These three sea routes represent more than half of the annual fleet tonnage and one-third of maritime traffic worldwide—this study systematically quantified and evaluated physical activity and sleep behavior among seafarers in Asian maritime settings. The main findings were that most participants showed deficient physical activity levels and poor sleep quality, regardless of whether their ship was sailing or moored. It is a novel and exciting study and may help to improve the condition of the seafarers.

Here are my comments

  1. The authors chose senior maritime students for their studies. It was not clear how many voyages were completed by the subjects before this study.
  2. Before initiating the study, the descriptive characters of the participants were recorded. I was curious if the authors recorded these characters (especially weight) after they made the voyage. Because a change in physical activity might increase weight. 
  3. Here authors always compared to sailing subjects to the moored subjects. It would be more informative if there was another control group of similar characters ( Height, weight, BMI etc.) who perform white-collar jobs on the shore/ land. 
  4. When recruiting the subjects for the study, did authors also consider any pre-existing medical condition of the subjects? Because any pre-existing condition may affect the study.
  5. The authors say that ship motion could explain the difference in sleep quality. I was curious, is there a method to measure the ship's motion when it is moored versus sailing?
  6. I must admit I have never traveled to any of the ships before. Therefore I wanted to know if there is any provision for doing the exercise (like treadmill, etc.) on the ship? Did the subjects were advised to do exercise?
  7. Although it is not possible in this study, I recommend that the authors consider testing testosterone and cortisol hormones and blood sugar in the test subjects before and after the voyage. It will enhance the qualify of the work and will give more insight.
  8. I agree with the authors that physical activity levels and sleep patterns should be examined under different seasonal sea conditions to make better conclusions.
  9. The authors do not mention temperature conditions on the ship while sailing versus moored. Temperature does have much effect on the physical activity of the subjects.
  10. The authors propose that worksite health promotion programs are required for those who spend long periods working aboard ships. It would be better if the authors recommend any programs or schedules for the seafarers based on this study.

Overall the study looks exciting and vital, and the manuscript is very well organized and written. I do understand recruiting 51 subjects for the study is not an easy task and I appreciate the effort, I recommend it for publications after they address the above questions.

Author Response

Reviewer's Notes:

  • The authors chose senior maritime students for their studies. It was not clear how many voyages were completed by the subjects before this study.

Authors’ Response: Thanks for your comment! The annual sailing plan of the training ship which was employed in the study contains sixteen voyages per year. The annual sailing plan consists of fourteen coastal voyages (i.e. sailing between Korean ports) and two international voyages. The experiment was conducted during the second semester, and experiment participants have experienced twelve domestic voyages and one international voyage before the experiment. Additional information has been added Line 90-94.

  • Before initiating the study, the descriptive characters of the participants were recorded. I was curious if the authors recorded these characters (especially weight) after they made the voyage. Because a change in physical activity might increase weight.

Authors’ Response: Good point! As the purpose of the study was focused on the cross-sectional comparison regarding physical activity level changes, the weight of participants was reported by the maritime students by themselves. As a reviewer pointed, physical changes including weight and blood pressure, etc., would be interesting variables to the understanding of seafarers’ physical changes before and after prolonged sea navigation, but unfortunately, we did not measure them.

  • Here authors always compared to sailing subjects to the moored subjects. It would be more informative if there was another control group of similar characters (Height, weight, BMI, etc.) who perform white-collar jobs on the shore/ land.

Authors’ Response: Thanks for sharing your research suggestion. Comparing the seafarers’ group with similar working groups such as port operators would be a good research goal for the next study. For this study, as the experiment of this study was a complex and relatively large scale, we focused on comparing onboard physical activity level and sleep.

  • When recruiting the subjects for the study, did the authors also consider any pre-existing medical condition of the subjects? Because any pre-existing condition may affect the study.

 Authors’ Response: As minimum requirements for onboard training, all trainee should officially report their medical condition to the government authority. Based on the medical condition report, it was confirmed that the participants for the study didn’t have any pre-existing medical condition as navigation officers. Additional information has been added Line 62-65.

  • The authors say that ship motion could explain the difference in sleep quality. I was curious, is there a method to measure the ship's motion when it is moored versus sailing?

Authors’ Response: The ship’s movement monitor is equipped for the training ship. The ship’s movement range between moored and sailing was clearly showing distinctive differences. It would be great if we can express as a number in the text.

  • I must admit I have never traveled to any of the ships before. Therefore I wanted to know if there is any provision for doing the exercise (like treadmill, etc.) on the ship? Did the subjects were advised to do exercise?

Authors’ Response: Yes, they were. The ship has a gymnasium with multiple exercise equipment for ship’s crews and maritime students to motivate their physical activity while sailing. Typically, ship’s crews are more frequently using the gym facility whereas maritime students are not preferred to use the gym facility, especially during sailing.

  • Although it is not possible in this study, I recommend that the authors consider testing testosterone and cortisol hormones and blood sugar in the test subjects before and after the voyage. It will enhance the quality of the work and will give more insight.

Authors’ Response: Good question! Similar to the comment No. 2, additional independent variables would enhance the quality of evaluation for mariners’ health while sailing, as recommended by the reviewer, physiological aspects of seafarers will be considered for the future study.

  • I agree with the authors that physical activity levels and sleep patterns should be examined under different seasonal sea conditions to make better conclusions.

Authors’ Response: Although the design of the experimental condition to compare different seasonal sea conditions is practically difficult, we will keep considering how the future study could admit the important aspect of sea condition with respect to the seafarers’ health.

  • The authors do not mention temperature conditions on the ship while sailing versus moored. Temperature does have much effect on the physical activity of the subjects.

Authors’ Response: Thanks for the good comment! The training ship is equipped with an air conditioner that sufficiently controls accommodation air temperature conditions. The air temperature of the inside of the accommodation is maintained a similar temperature in both mooring and sailing conditions.

  • The authors propose that worksite health promotion programs are required for those who spend long periods working aboard ships. It would be better if the authors recommend any programs or schedules for the seafarers based on this study.

Authors’ Response: Ship’s facility and working conditions are affected by international agreements such as the World Health Organization (WHO) and the International Maritime Organization (IMO). In order to specify the health promotion programs, we are currently investigating various international regulations and guidelines. Based on the results of the study, the proposal of the mariner’s health promotion will be conducted as another full-scale research.

Overall the study looks exciting and vital, and the manuscript is very well organized and written. I do understand recruiting 51 subjects for the study is not an easy task and I appreciate the effort, I recommend it for publications after they address the above questions.

Authors’ Response: Thank you very much for your comments and recommendation!

Best regards,

Reviewer 3 Report

Prolonged ocean voyages constrain the regular physical activity and sleep patterns of seafarers. Here authors examined physical activity and sleep patterns among seafarers using a single wrist-worn accelerometer which is different with previous studies. Their study cohort included fifty-one senior maritime students. Three conditions were compared: 1. mooring&sailing; 2. On-duty & off-duty; 3. Day & night navigation duty. Their main finding is that physical activity levels and sleep quality of seafarers were considerably lower than the guidelines.

Overall, this is an interesting research. Here are my comments:

Major Comments:

  1. Author selected sea routes in the Pacific region. Whether different marine environments will also affect the results. So, I think that authors can choose a different route to do the next study.
  2. Authors studied the effects of short-term sailing on physical activity and sleep. But long-term changes in sleep have a significant impact on physical health. Only long-term changes in sleep have a significant impact on physical health. Authors should analyze long-term sailing on physical activity and sleep.
  3. Does prolonged ocean voyages affect behavior and mood?

Author Response

Reviewer's Comments:

  • Author selected sea routes in the Pacific region. Whether different marine environments will also affect the results. So, I think that authors can choose a different route to do the next study.

Authors’ Response: Thanks for your comment! The authors agree with the reviewer’s comment. The next study will investigate the seasonal changes of the same sea route. Additionally, the Northern Pacific route would be another interesting sea route that may differently affect the seafarers’ health while sailing.

  • Authors studied the effects of short-term sailing on physical activity and sleep. But long-term changes in sleep have a significant impact on physical health. Only long-term changes in sleep have a significant impact on physical health. Authors should analyze long-term sailing on physical activity and sleep.

Authors’ Response: As the reviewer pointed, seafarers’ health in the more realistic sea working environment would provide a better understanding of mariners’ health changes at sea. Although the design of the experimental configuration and arrange a ship participating in a long-distance sailing would be practically difficult, we will keep considering how the future study could admit the important aspect of sea condition with respect to the seafarers’ health.

  • Does prolonged ocean voyages affect behavior and mood?

Authors’ Response: We guess so. Based on the relevant literature (Xia, Song-Yun, et al. "Psychological and physiological effects of a long voyage on female seamen in China." Int J Clin Exp Med 9.4 (2016): 7314-7321.), it was revealed that chronic psychological stress during a long voyage effect on the mariner’s psychological and physiological behavior. As our study was intended to compare the physical aspect of the mariner’s health, the submitted paper was not mentioned about the effect of behavior and mood.

Best regards,